# Chemically tailored block copolymers for highly reliable sub-10-nm patterns by directed self-assembly

Shinsuke Maekawa ®[1], Takehiro Seshimo[2], Takahiro Dazai[2], Kazufumi Sato[2], Kan Hatakeyama-Sato[1], Yuta Nabae ®[1] & Teruaki Hayakawa ®[1] ✉

While block copolymer (BCP) lithography is theoretically capable of printing features smaller than 10 nm, developing practical BCPs for this purpose remains challenging. Herein, we report the creation of a chemically tailored, highly reliable, and practically applicable block copolymer and sub-10-nm line patterns by directed self-assembly. Polystyrene-*block*-[poly(glycidyl methacrylate)-*random*-poly(methyl methacrylate)] (PS-*b*-(PGMA-*r*-PMMA) or PS-*b*-PGM), which is based on PS-*b*-PMMA with an appropriate amount of introduced PGMA (10–33 mol%) is quantitatively post-functionalized with thiols. The use of 2,2,2-trifluoroethanethiol leads to polymers (PS-*b*-PG$_F$Ms) with Flory–Huggins interaction parameters ($\chi$) that are 3.5–4.6-times higher than that of PS-*b*-PMMA and well-defined higher-order structures with domain spacings of less than 20 nm. This study leads to the smallest perpendicular lamellar domain size of 12.3 nm. Furthermore, thin-film lamellar domain alignment and vertical orientation are highly reliably and reproducibly obtained by directed self-assembly to yield line patterns that correspond to a 7.6 nm half-pitch size.

As the performance of electronic devices and artificial intelligence systems advances rapidly, the need for increasingly finer circuit patterns on semiconductor chips becomes more critical. By 2037, it is projected that the required minimum half-pitch for 0.5 nm logic nodes will be as small as 8 nm, highlighting a pressing need for advancements in lithographic processes[1]. In response to this challenge, the integration of directed self-assembly (DSA) of block copolymers (BCPs) with extreme ultraviolet (EUV) lithography emerges as a promising candidate of next-generation lithography. This combination has a potential to reduce the pitch-size with reducing line edge roughness.

Assembled microphase-separated BCPs are attracting increasing levels of interest due to their potential use in batteries, solar cells, fuel cells, porous materials, and next-generation lithography for the semiconductor industry[2–5]. Perpendicularly oriented microphase-separated structures in BCP thin films can act as templates for pattern transfer to any underlying substrate[6–8]. The morphologies and

sizes of these nano-assemblies depend on the volume fractions of the constituent polymers, levels of incompatibility between blocks (Flory–Huggins parameter, $\chi$), and degrees of polymerization ($N$)[9–11]. Over the past few decades, polystyrene-*block*-poly(methyl methacrylate) (PS-*b*-PMMA) has received the most academic and industrial attention for use in DSA applications[12–15] for the following three reasons: (1) PS and PMMA have almost identical surface free energies (SFEs) at the thermal-annealing temperature[16], (2) random copolymers are available for controlling interfacial interactions[17–22], and (3) the selective removal of PMMA domains facilitates easy pattern transfer[23–27]. Almost identical SFEs of the blocks are necessary for perpendicular orientation of microphase-separated structures in BCP thin films. Unfortunately, the smallest lithographically useful features formed in PS-*b*-PMMA are approximately 11 nm wide due to thermodynamically driven block mixing when shorter BCP chains are used to produce smaller features[28,29]. In theory, BCPs with $\chi N$ values of more

[1]Department of Materials Science and Engineering, School of Materials and Chemical Technology, Tokyo Institute of Technology, Tokyo 152-8552, Japan. [2]Research & Development Department, Tokyo Ohka Kogyo Co., Ltd., Kanagawa 253-0114, Japan. ✉e-mail: hayakawa.t.ac@m.titech.ac.jp

than 10.5 can potentially be used to produce microphase-separated structures[9], and BCPs with both low $N$ and high $\chi$ values have been studied in order to achieve microphase-separated structures with periodic lengths of less than 20 nm. High-$\chi$ BCP systems are typically designed by pairing polymers with very different chemical properties, such as hydrophobic–hydrophilic[30,31], organic–inorganic[32,33], or fluorine-containing BCPs[34,35]. However, the segment with the lower SFE selectively segregates to the surface, with parallel-oriented structures formed in most of the high-$\chi$ BCP thin films.

Several approaches have been established to promote perpendicularly oriented domains in high-$\chi$ materials. A typical approach aims to achieve segments with equal interfacial energies by introducing a neutral top-coat layer that replaces the free air surface[36–38]. Willson et al. reported 5 nm features by a DSA in poly(5-vinyl-1,3-benzodioxole-*block*-pentamethyldisilylstyrene) (PVBD-*b*-PDSS) thin films with a top-coat[39]. Solvent-vapor annealing is another approach[40–42] in which microphases separate in the solvated layer, which neutralizes the interfacial energy of the segments at the top of the thin film. Fleury et al. reported perpendicular lamellae with feature size of 7.3 nm in polystyrene-*block*-poly(2-fluoroethylmethyl acrylate) (PS-*b*-P2FEMA) by solvothermal annealing[43]. However, top-coat and solvent-vapor annealing have issues of higher-cost and low level of safety, respectively. An alternative strategy involves designing a material with a higher-$\chi$ than PS-*b*-PMMA and blocks with similar SFEs that promote the perpendicular orientation during thermal annealing; such materials include PS-*b*-PDLA ($L_0 = 15.9$ nm)[44], PS-*b*-PPC ($L_0 = 16.8$ nm)[45], PS-*b*-PI with epoxidized PI ($L_0 = 14.6$ nm)[46,47], PSVN-*b*-PMMA ($L_0 = 20$ nm)[48], ester–amide PS-*b*-PMMA exchanged with ethanolamine ($L_0 = 17$ nm)[49], P(S-*gradient*-PFS)-*b*-PMMA ($L_0 = 18.2$ nm)[50], and A-*b*-(B-*r*-C) type polystyrene-*block*-poly(glycidyl methacrylate) (PS-*b*-PGMA) derivatives functionalized with two different thiols ($L_0 = 8.0$ nm)[51]. We previously reported, polystyrene-*block*-poly[2-hydroxy-3-(2,2,2-trifluoroethylsulfanyl)-propyl methacrylate] (PS-*b*-PHFMA), an original high-$\chi$ BCP synthesized from PS-*b*-PGMA and 2,2,2-trifluoroethanethiol[52]. PS-*b*-PHFMA has an effective $\chi$ ($\chi_{eff}$) value of 0.167 at 200 °C and a bulk state with a 9.6-nm lamellar structure. In addition, the hydrophobic 2,2,2-trifluoroethyl groups oppose the hydroxy groups in the PHFMA blocks, which balances the SFEs of the two PS-*b*-PHFMA segments and facilitates the formation of perpendicular lamellae with $L_0$ of 26 nm in thin films during thermal annealing. While some of these pioneering studies have created perpendicular lamellae by thermal annealing, creating sub-10 nm features by DSA in absence of a top-coat remains challenging.

In this study we introduce 2,2,2-trifluoroethyl groups into the PMMA segments of PS-*b*-PMMA while maintaining precise control over the chemical modification rate in order to increase $\chi_{eff}$ without radically changing the desirable surface and interfacial properties of PS-*b*-PMMA. We design two PS-*b*-PMMA derivatives with higher $\chi$ values based on this concept, namely polystyrene-*block*-[poly(glycidyl methacrylate)-*random*-poly(methyl methacrylate)] (PS-*b*-(PGMA-*r*-PMMA) or PS-*b*-PGM) with small amounts of PGMA (≤ 30 mol% PGMA in the PGM segments) and PS-*b*-(PGMA$_F$-*r*-PMMA) (PS-*b*-PG$_F$M) modified with 2,2,2-trifluoroethyl groups by referring to PS-*b*-PHFMA[52] (Fig. 1). PS-*b*-PG$_F$M consists of moderately hydrophobic PS blocks and PG$_F$M blocks containing hydrophilic carboxylic esters, hydroxy moieties, and hydrophobic 2,2,2-trifluoroethyl groups. We optimize the composition of the random PG$_F$M block in PS-*b*-PG$_F$M to increase $\chi_{eff}$ and simultaneously balance the surface affinities of the blocks. Small-angle X-ray scattering (SAXS) analysis is used to determine morphology, domain size (*d*-spacing), and $\chi_{eff}$. PS-*b*-PG$_F$M thin films form well-ordered perpendicular lamellae, as characterized by atomic force microscopy (AFM) and scanning electron microscopy (SEM). The PS-*b*-PG$_F$M polymers are also used in the chemo-epitaxial DSA process previously optimized for PS-*b*-PMMA.

## Results

### PS-*b*-PGM synthesis and post-functionalization

A series of PS-*b*-PGMs was successfully synthesized by the sequential living anionic polymerization of styrene and a mixture of glycidyl methacrylate (GMA) and methyl methacrylate (MMA). The reaction was initiated by *sec*-butyllithium (*sec*-BuLi) in the presence of excess lithium chloride (LiCl) and 1,1-diphenylethylene (DPE) in tetrahydrofuran (THF) at −78 °C under argon (Fig. 2)[53–55]. The synthesized PS-*b*-PGMs were purified by Soxhlet extraction using *n*-hexane, cyclohexane, or an *n*-hexane/cyclohexane mixture to remove small amounts of PS homopolymer formed due to water (as a contaminant) in the reaction vessel during the polymerization process. The volume fraction of PS ($f_{PS}$) and the PGMA content in the PGM segment were determined by ${}^1$H NMR spectral integration (Table 1), assuming densities of 1.04, 0.805, and 1.18 g cm${}^{-3}$ for PS, PGMA, and PMMA, respectively. The number-averaged molecular weights ($M_n$) and dispersities ($Đ$) of the PS-*b*-PGMs were determined by size-exclusion chromatography (SEC) against PS standards using THF as the eluent. $M_n$ values were maintained close to 20, 10, and 5 kg mol${}^{-1}$. PS-*b*-PGMs with various molecular weights ($X$ kg mol${}^{-1}$) were synthesized with appropriate PGMA contents ($Y = 30$, 20, and 10 mol% in the PGM segment) to control the affinities of the segments toward air, with products referred to as "PS-*b*-PGM$X$-$Y$".

PS-*b*-PGMs were functionalized using LiOH-catalyzed epoxide-ring-opening chemistry[56–59] (thiol–epoxy reaction) with excess 2,2,2-trifluoroethanethiol, ethanethiol, benzenethiol, 2-phenylethanethiol, or cyclohexanethiol in THF, to clarify how repulsions between BCP segments are affected by the introduced functional groups; these products are referred to as "PS-*b*-PG$_F$M", "PS-*b*-PG$_H$M", "PS-*b*-PG$_{Ph}$M", "PS-*b*-PG$_{C2Ph}$M", and "PS-*b*-PG$_{Cy}$M", respectively (Fig. 2). PGMA-unit conversions were determined by integrating the glycidyl peaks in the ${}^1$H NMR spectra (4.26–4.38, 3.85–3.75, 3.23, 2.86, and 2.64 ppm) (Table 2). The functionalized polymers exhibited lower integrated glycidyl-peak intensities compared to the PS-*b*-PGM precursors. In addition, new signals derived from the introduced functional groups suggest that the epoxides in the PGMA segments had been ring-opened by the thiols. While most PS-*b*-PGMs reacted quantitatively with 2,2,2-trifluoroethanethiol, PS-*b*-PGM19-10 and PS-*b*-PGM18-11 reacted with PGMA conversions of 96% and 92%, respectively (Supplementary Figs. 20 and 21), which was ascribable to low concentrations of PGMA units in the reaction mixtures at $Y ≤ 11$ mol%. Therefore, the resultant two polymers were re-reacted under the same conditions to consume unreacted PGMA segments and afford PS-*b*-PG$_F$M19-10 and PS-*b*-PG$_F$M18-11. A few ${}^1$H NMR peaks that correspond to the glycidyl groups of PGMA were observed in the spectra of PS-*b*-PG$_H$M10-22, PS-*b*-PG$_{C2Ph}$M19-23, PS-*b*-PG$_{Cy}$M19-23, and PS-*b*-PG$_{Cy}$M10-22, which were ascribable to variations in thiol nucleophilicity. ${}^1$H NMR spectroscopy and density data (1.04, 1.18, 1.43, 1.23, 1.27, 1.21, and 1.18 g cm${}^{-3}$ for PS, PMMA, PGMA$_F$[52], PGMA$_H$[52], PGMA$_{Ph}$, PGMA$_{C2Ph}$, and PGMA$_{Cy}$, respectively) were used to calculate $f_{PS}$ values for the various PS-*b*-PG$_R$Ms. Each synthesized PS-*b*-PG$_R$M exhibited a unimodal, symmetrical, and narrowly dispersed SEC trace (Supplementary Fig. 22), consistent with the thiol–epoxy reactions proceeding without any competing side reactions. The NMR and SEC data reveal that the PS-*b*-PGM precursors and their derivatives were successfully synthesized by combining sequential living anionic polymerization with the thiol–epoxy reaction.

### Bulk morphologies

SAXS and transmission electron microscopy (TEM) were used to analyze the morphologies and determine the *d*-spacings of the microphase-separated structures formed in bulk films of the synthesized BCPs, which were prepared by the slow evaporation of BCP solutions in THF followed by thermal annealing at reduced pressure and 200 °C for 24 h. SAXS profiles were obtained at room temperature

at reduced pressure, and *d*-spacings were determined from the positions (*q**) of first-order scattering peaks (*d*-spacing = $2\pi/q^*$). Thin sections of bulk films were prepared by ultramicrotomy, stained with ruthenium tetroxide (RuO$_4$), and then observed by TEM.

We studied bulk PS-*b*-PGM19-23, PS-*b*-PGM10-22, and their derivatives to clarify how the introduced functional groups affect the morphology of each microphase-separated structure and its *d*-spacing (Fig. 3a–c). The SAXS profiles of PS-*b*-PGM19-23 and PS-*b*-PG$_R$M19-23 exhibited peaks at integer ratios relative to their first-order peaks, which suggests that lamellar structures had formed. PS-*b*-PG$_F$M19-23 had a *d*-spacing of 19.5 nm, which was larger than that of its precursor (16.5 nm). In contrast, PS-*b*-PG$_{C2Ph}$M19-23 and PS-*b*-PG$_{Cy}$M19-23 exhibited lower *d*-spacings of 16.2 and 15.0 nm, respectively. Because the PS-*b*-PG$_R$M19-23 BCPs are all derived from the same precursor, their molecular chains should all be the same length; consequently, their *d*-spacings only depend on the interfacial thicknesses of their microphase-separated structures[60]. Functionalized PS-*b*-PG$_F$M19-23 exhibited a larger *d*-spacing suggesting narrower interfacial area compared to the PS-*b*-PGM19-23 precursor owing to higher inter-segment repulsion. In contrast, PS-*b*-PG$_{C2Ph}$M19-23 and PS-*b*-PG$_{Cy}$M19-23 exhibited the reverse trend because the introduced functional groups promoted segment mixing. PS-*b*-PGM10-22 (with a molecular weight of 10 kg mol$^{-1}$) displayed a disordered structure while its PS-*b*-PG$_F$M10-22 derivative formed a lamellar structure with a *d*-spacing of

12.4 nm (Fig. 3d, e, respectively), which suggests that the $\chi N$ value of PS-*b*-PGM10-22 was less than 10.5 and introducing the trifluoroethyl group is a suitable method for increasing the $\chi$ value of the BCP to form finer phase-separated structures.

The morphologies of the synthesized PS-*b*-PG$_F$Ms were analyzed to investigate the relationship between the primary and higher-order structures of the BCPs. With the exception of PS-*b*-PG$_F$M10-11, the SAXS profiles of PS-*b*-PG$_F$Ms with $M_n \geq 10$ kg mol$^{-1}$ exhibited sharp peaks at integer ratios relative to their first-order peaks, suggestive of the formation of well-ordered lamellar structures. In contrast, the SAXS profile of PS-*b*-PG$_F$M10-11 ($M_n = 11$ kg mol$^{-1}$, PGMA content = 11 mol%) exhibited a single broad first-order peak indicative of a disordered structure, which implies that inter-segment repulsion strengthens with increasing PGMA content in the PG$_F$M segment. Bulk BCP films with low molecular weights ($M_n = 5$ kg mol$^{-1}$) also formed disordered structures. Thus, introducing 10 mol% trifluoroethyl groups into the PMMA block of PS-*b*-PMMA is expected to produce templates with the required properties for use in next-generation lithography.

### Determining the effective Flory–Huggins interaction parameter ($\chi_{eff}$)

The segregation strengths of PG$_F$Ms with various PGMA$_F$ contents relative to PS were quantified using the random-phase approximation method[52,61–64]. Specifically, segregation strength can be described by the effective Flory–Huggins interaction parameter $\chi_{eff}$ (= $\alpha + \beta/T$, where $\alpha$ is the entropic contribution and $\beta/T$ is the enthalpic contribution). This method requires SAXS data to be acquired above the mean-field crossover temperature to avoid the effects of thermal fluctuation. PS-*b*-PG$_F$M5-10, PS-*b*-PG$_F$M5-22, and PS-*b*-PG$_F$M5-30 (all of which formed disordered structures at room temperature) were first subjected to temperature-dependent SAXS in 10 °C decrements starting from 230, 290, and 290 °C, respectively. $I_{max}^{-1}$–$T^{-1}$ plots (Supplementary Fig. 38) show discontinuous changes that are attributable to mean-field to non-mean-field transitions of disordered states at 150–160, 160–170, and 180–190 °C for PS-*b*-PG$_F$M5-10, PS-*b*-PG$_F$M5-22, and PS-*b*-PG$_F$M5-30, respectively. The SAXS profile of each mean-field-disordered state was further analyzed using Leibler's mean-field theory modified to include the effects of molecular weight dispersity and asymmetry in the segmental volume (Supplementary Fig. 39)[52,61–64]. A common reference volume ($v_0 = 118$ Å$^3$) was used to compare the values of $\chi_{eff}$ obtained with other reported values because $\chi_{eff}$ depends on the reference volume used to calculate $N$. Figure 4a shows that $\chi_{eff}$ was well-fitted to

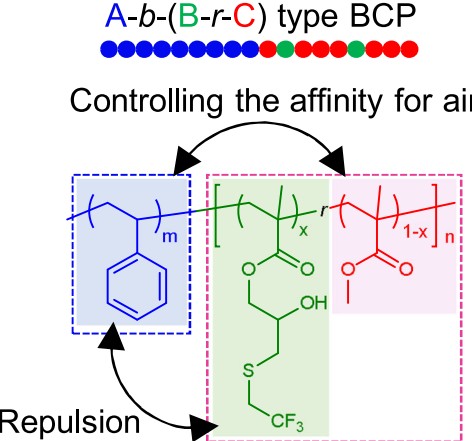

**Fig. 1 | Concept and molecular structure of the target BCP.**

**Fig. 2 | Scheme depicting the synthesis of PS-*b*-(PGMA$_R$-*r*-PMMA) (PS-*b*-PG$_R$M) by living anionic polymerization and post-functionalization involving the thiol–epoxy reaction.**

**Table 1 | Characterization data for synthesized PS-*b*-PGMs**

| Sample[a] | $M_n$[b] (kg mol⁻¹) | Đ[b] | $f_{PS}$[c] | PGMA[c] (mol%) | Morphology[d] | *d*-spacing[e] (nm) |
|---|---|---|---|---|---|---|
| PS-*b*-PGM20-33 | 20.2 | 1.04 | 0.513 | 33 | Lamellar | 15.2 |
| PS-*b*-PGM19-23 | 18.7 | 1.03 | 0.463 | 23 | Lamellar | 16.5 |
| PS-*b*-PGM19-10 | 19.0 | 1.03 | 0.499 | 10 | Disordered | – |
| PS-*b*-PGM18-11 | 17.5 | 1.04 | 0.476 | 11 | Disordered | – |
| PS-*b*-PGM10-33 | 9.63 | 1.06 | 0.504 | 33 | Disordered | – |
| PS-*b*-PGM10-22 | 10.1 | 1.08 | 0.532 | 22 | Disordered | – |
| PS-*b*-PGM10-11 | 10.4 | 1.09 | 0.532 | 11 | Disordered | – |
| PS-*b*-PGM5-30 | 4.83 | 1.12 | 0.529 | 30 | Disordered | – |
| PS-*b*-PGM5-22 | 4.76 | 1.10 | 0.500 | 22 | Disordered | – |
| PS-*b*-PGM5-10 | 5.20 | 1.14 | 0.500 | 10 | Disordered | – |

[a]PS-*b*-PGM*X-Y* refers to a BCP with $M_n = X$ kg mol⁻¹ and with *Y* mol% PGMA units in its PGM segment.
[b]Determined by SEC in THF against PS standards.
[c]Determined by ¹H NMR spectroscopy in CDCl₃.
[d]Determined by SAXS peak ratios and TEM.
[e]Determined by the position of the first-order peak in the SAXS profile.

**Table 2 | Characterization data for synthesized PS-*b*-PG$_R$Ms**

| Sample[a] | $M_n$[b] (kg mol⁻¹) | Đ[b] | $f_{PS}$[c] | PGMA conversion[c] | Morphology[d] | *d*-Spacing[e] (nm) | $L_o$[f] (nm) |
|---|---|---|---|---|---|---|---|
| PS-*b*-PG$_F$M20-33 | 19.3 | 1.04 | 0.520 | >99% | Lamellar | 19.5 | 18.6 |
| PS-*b*-PG$_F$M19-23 | 20.8 | 1.03 | 0.471 | >99% | Lamellar | 19.5 | 18.5 |
| PS-*b*-PG$_F$M19-10 | 21.0 | 1.03 | 0.497 | >99% | Lamellar | 18.1 | 16.8 |
| PS-*b*-PG$_F$M18-11 | 17.1 | 1.04 | 0.514 | >99% | Lamellar | 15.6 | 15.1 |
| PS-*b*-PG$_F$M10-33 | 12.0 | 1.05 | 0.510 | >99% | Lamellar | 13.2 | 12.5 |
| PS-*b*-PG$_F$M10-22 | 10.0 | 1.09 | 0.582 | >99% | Lamellar | 12.4 | 12.3 |
| PS-*b*-PG$_F$M10-11 | 11.1 | 1.08 | 0.532 | >99% | Disordered | – | – |
| PS-*b*-PG$_F$M5-30 | 6.02 | 1.09 | 0.570 | >99% | Disordered | – | – |
| PS-*b*-PG$_F$M5-22 | 6.06 | 1.08 | 0.548 | >99% | Disordered | – | – |
| PS-*b*-PG$_F$M5-10 | 5.39 | 1.13 | 0.475 | >99% | Disordered | – | – |
| PS-*b*-PG$_H$M19-23 | 21.3 | 1.03 | 0.481 | >99% | Lamellar | 16.4 | – |
| PS-*b*-PG$_H$M10-22 | 9.67 | 1.09 | 0.591 | 96% | Disordered | – | – |
| PS-*b*-PG$_{Ph}$M19-23 | 21.7 | 1.03 | 0.455 | >99% | Lamellar | 16.4 | – |
| PS-*b*-PG$_{Ph}$M10-22 | 10.1 | 1.07 | 0.566 | >99% | Disordered | – | – |
| PS-*b*-PG$_{C2Ph}$M19-23 | 20.0 | 1.03 | 0.449 | 98% | Lamellar | 16.2 | – |
| PS-*b*-PG$_{C2Ph}$M10-22 | 11.1 | 1.09 | 0.550 | >99% | Disordered | – | – |
| PS-*b*-PG$_{Cy}$M19-23 | 19.9 | 1.03 | 0.472 | 89% | Lamellar | 15.0 | – |
| PS-*b*-PG$_{Cy}$M10-22 | 10.8 | 1.08 | 0.578 | 96% | Disordered | – | – |

[a]PS-*b*-PG$_R$M*X-Y* was synthesized from the corresponding PS-*b*-PGM*X-Y,* and "$_R$" designate each thiol.
[b]Determined by SEC in THF against PS standards.
[c]Determined by ¹H NMR spectroscopy in CDCl₃.
[d]Determined by SAXS peak ratios and TEM.
[e]Determined by the position of the first-order peak in the SAXS profile.
[f]Determined from the AFM phase image.

$\chi_{eff} = \alpha + \beta/T$ for each sample; fitted values of $\alpha$ and $\beta$ are listed in Supplementary Table 9. PS-*b*-PG$_F$M5-10, PS-*b*-PG$_F$M5-22, and PS-*b*-PG$_F$M5-30 were determined to have $\chi_{eff}$ values of 0.110, 0.133, and 0.142, respectively, at 200 °C. Interestingly, $\chi_{eff}$ increased with increasing PGMA$_F$ content for the PS-*b*-PG$_F$M series at 200 °C (Fig. 4b); PS-*b*-PG$_F$M with a PGMA$_F$ content of 10 mol% exhibited a $\chi_{eff}$ value 3.5-times that of PS-*b*-PMMA ($\chi_{eff} = 0.031$[65]).

The sudden rise in the $\chi_{eff}$ value of the PS-*b*-PMMA derivative could be attributed to the introduction of specific functional groups within its structure. The presence of hydroxy and trifluoroethyl groups in 10–30 mol% concentrations in PMMA seems to enhance the $\chi_{eff}$ value significantly. This increase is likely due to enhanced interactions within the polymer blocks, similar to the observed self-interactions in PSVN-*b*-PMMA caused by vinylnaphthalene units[48] and complementary hydrogen bonding between adenine and thymine moieties in P(VBA-*r*-VBT)-*b*-PS[66]. In this study, hydrogen bonding by hydroxy groups introduced into the PMMA block increased intra-block interactions within PG$_F$M. In addition, the $\chi_{eff}$ values of A-*b*-(B-*r*-C) copolymers are estimated by ref. 67,68

$$\chi_{eff} = x\chi_{AB} + (1-x)\chi_{AC} - x(1-x)\chi_{BC} \tag{1}$$

where $\chi_{AB}$, $\chi_{AC}$, and $\chi_{BC}$ represent $\chi$ parameter between A and B, A and C, and B and C, respectively, and $x$ indicates the fraction of B in B-*r*-C block. In the case of $\chi_{AB} > \chi_{AC}$ and $(\chi_{AB}-\chi_{AC}) > \chi_{BC}$, the $\chi_{eff}$ value increases with increasing $x$[49]. The $\chi$ values between constituent

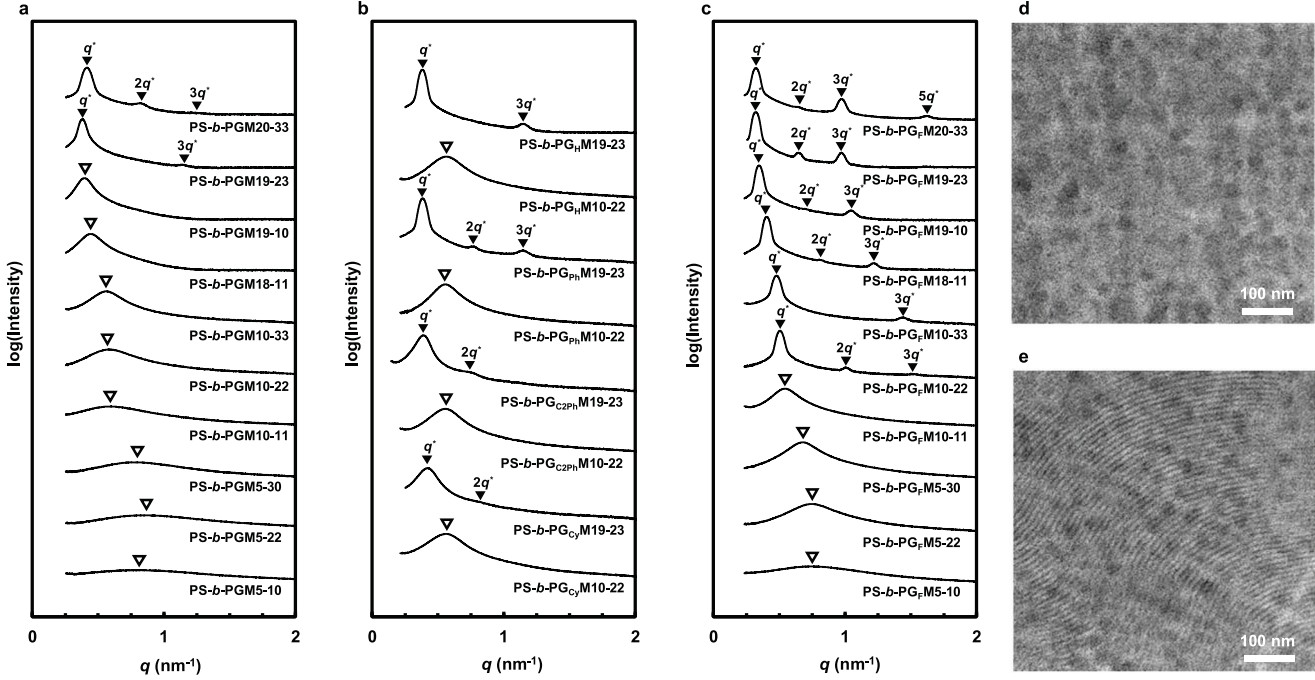

**Fig. 3 | Bulk properties of the synthesized BCPs.** SAXS profiles of bulk **a** PS-*b*-PGM, **b** PS-*b*-PG$_R$M, and **c** PS-*b*-PG$_F$M samples. TEM images of **d** PS-*b*-PGM10-22 and **e** PS-*b*-PG$_F$M10-22. The dark regions correspond to the PS blocks stained with RuO$_4$.

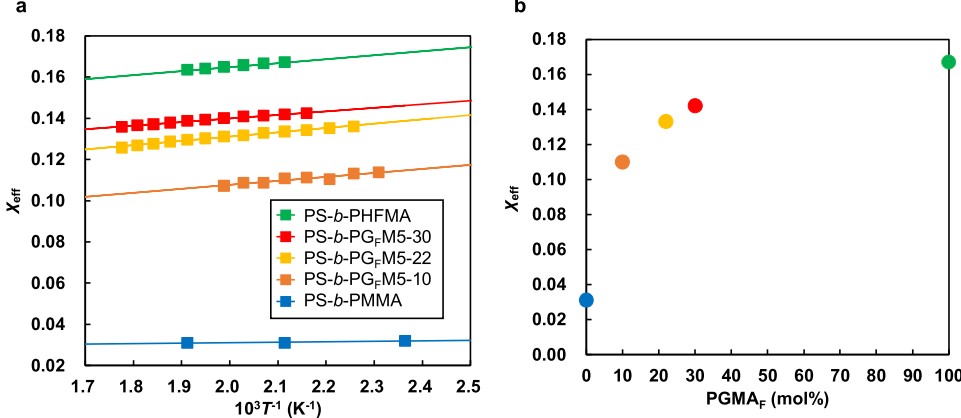

**Fig. 4 | Relationship between $\chi_{eff}$ and PGMA$_F$ content. a** $\chi_{eff}$ values of PS-*b*-PMMA[65], PS-*b*-PG$_F$M5-10, PS-*b*-PG$_F$M5-22, PS-*b*-PG$_F$M5-30, and PS-*b*-PHFMA[52] as functions of temperature with a reference volume of 118 Å³. **b** $\chi_{eff}$ of PS-*b*-PMMA[65], PS-*b*-PG$_F$Ms, and PS-*b*-PHFMA[52] at 200 °C as a function of the PGMA$_F$ content in the PG$_F$M segment. In this study, the instrumental broadening factor was not measured. However, the $\chi_{eff}$ values of PS-*b*-PMMA[65] estimated using the same SAXS

instrument were in good agreement with the $\chi_{eff}$ values of earlier studies[28,70]. In addition, the SAXS profiles used in this study (Supplementary Fig. 39) were broadened, so the effect of the instrumental broadening factor to the estimated $\chi_{eff}$ values was considered to be small. Further discussion on the instrumental broadening factor and estimated $\chi_{eff}$ values is described in the Supplementary Discussion.

homopolymers of PS-*b*-PG$_F$M have been reported[58] and meet the conditions. Based on the reasons, the $\chi_{eff}$ value of PS-*b*-PG$_F$M increases with increasing PGMA$_F$ content and the parabolic trend has been shown.

## Thin-film study

In order to obtain ultrafine perpendicular lamellar structures with lamellar domains ($L_0$) of less than 20 nm, PS-*b*-PG$_F$M (which form lamellar structures in their bulk states) thin films were fabricated on silicon substrates that were grafted with polystyrene-*random*-poly(methyl methacrylate)-*random*-poly(2-hydroxyethyl methacrylate) (PS-*r*-PMMA-*r*-PHEMA) to neutralize polymer/substrate interactions involving the segments. A series of PS-*r*-PMMA-*r*-PHEMA polymers were synthesized by the free-radical polymerization of styrene, MMA, and

2-hydroxyethylmethacrylate (HEMA) with 2,2'-azobis(isobutyronitrile) (AIBN) as the initiator (Supplementary Table 11). The compositions of the various substrate-grafted random copolymers and the thickness of each BCP thin film are summarized in Supplementary Tables 11 and 12. Figure 5 shows AFM phase images of PS-*b*-PG$_F$M thin films prepared by spin-coating and thermal annealing. The observed fingerprint-like patterns suggest the formation of perpendicular lamellar structures on the surface-modified substrates. However, parallel lamellae were also observed on the same substrates as the perpendicular lamellae in the PS-*b*-PG$_F$M20-33 and PS-*b*-PG$_F$M10-33 thin films (Fig. 5c, f). These results suggest that the segments in PS-*b*-PG$_F$Ms with PGMA contents of 23 mol% or less displayed balanced air affinities. The surface free energies (SFEs) determined for the PG$_F$Ms also suggests that such affinities were balanced (Supplementary Table 6). In particular,

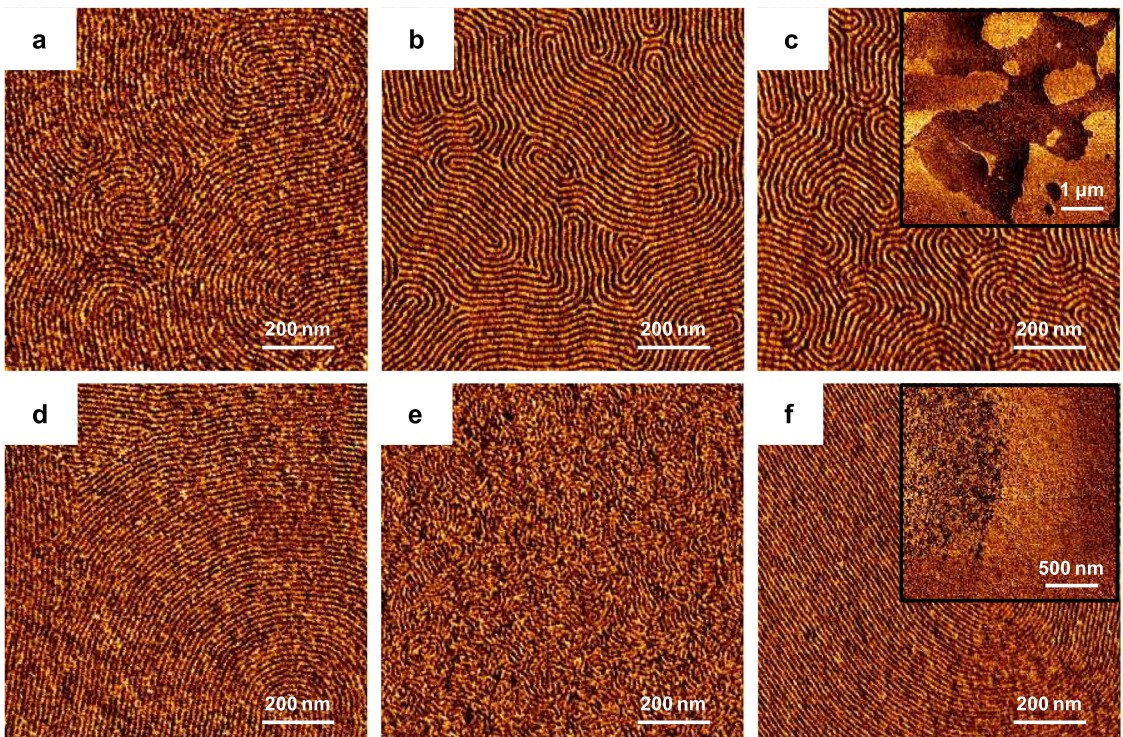

**Fig. 5 | Orientation control of the lamellae in PS-*b*-PG_FM thin films.** AFM phase images of **a** PS-*b*-PG_FM19-10, **b** PS-*b*-PG_FM19-23, **c** PS-*b*-PG_FM20-33, **d** PS-*b*-PG_FM18-11, **e** PS-*b*-PG_FM10-22, and **f** PS-*b*-PG_FM10-33 thin films prepared on chemically modified Si wafers and thermally annealed at 200 °C for 30 min. The insets in **c** and **f** show that both parallel and perpendicular orientations also formed on the same substrate in these cases.

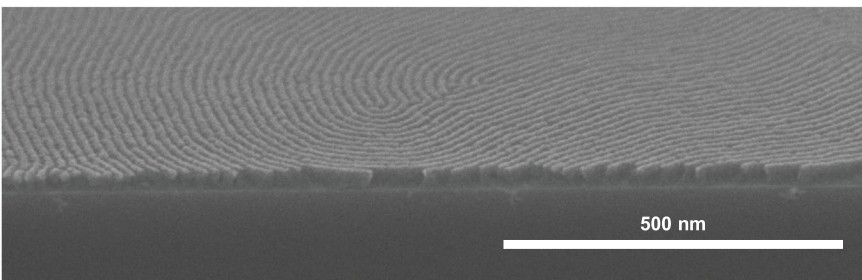

**Fig. 6 | Tilted SEM image of a PS-*b*-PG_FM19-10 thin film.** The 21.1-nm-thick thin film on a Si substrate modified with a PS-*r*-PMMA-*r*-PHEMA thin film with a PS molar ratio of 50% (NL50) was annealed at 200 °C for 30 min under nitrogen. Dark regions correspond to PS blocks stained with RuO₄.

PG_FM9-12 ($M_n$ = 11 kg mol$^{-1}$, PGMA content = 12 mol%) was determined to have an SFE of 40.0 mJ m$^{-2}$. It should be noted that perpendicular lamellae were obtained using high-$\chi$ BCPs ($\chi_{eff}$ = 0.110–0.142) in this study, whereas perpendicular lamellae are typically achieved with low-$\chi$ BCPs, such as PS-*b*-PMMA ($\chi_{eff}$ = 0.031[65]). The perpendicular lamellae were found to have a minimum $L_0$ of 12.3 nm, which is smaller than the minimum dimension achievable by PS-*b*-PMMA (full pitch = 17.5 nm[69]).

SEM images of a tilted PS-*b*-PG_FM19-10 thin film were acquired to examine its internal structure and confirm the formation of perpendicular lamellae. The film was annealed at 200 °C for 30 min, stained with RuO₄, and then sectioned by cutting in liquid nitrogen. The cross-sectional image in Fig. 6 reveals that lamellar domains extend from the surface to the substrate, which confirms that perpendicular lamellae had formed in this thin film.

**Directed self-assembly**
We finally examined the DSA compatibilities of PS-*b*-PG_FMs for producing large-area line patterns in thin films on surface-modified Si substrates using three representative polymers (PS-*b*-PG_FM19-23, PS-*b*-PG_FM19-10, and PS-*b*-PG_FM18-11). Si wafers modified with silicon nitride (SiN) layer and PS chemical guides were used as DSA substrates, which were further modified with PS-*r*-PMMA-*r*-PHEMA random copolymers (Fig. 7a). Thin PS-*b*-PG_FMs films were applied onto the substrates and annealed at 230 or 240 °C for 5 min. Figure 7b–d shows that DSA with 5× or 6× density multiplications were achieved using these PS-*b*-PG_FMs on PS chemical patterns with a pitch ($L_s$) of 84 or 90 nm. PS-*b*-PG_FM19-23 and PS-*b*-PG_FM18-11 line patterns self-assembled with 5× or 6× multiplications, respectively, over large areas on DSA substrates when $L_s$ = 90 nm (Fig. 7b, d), while a PS-*b*-PG_FM19-10 line pattern was obtained with a 5× density multiplication over a large area when $L_s$ = 84 nm (Fig. 7c). However, structural defects were observed in the PS-*b*-PG_FM19-23 thin film (Supplementary Fig. 40). The smallest feature was 7.6 nm in size; hence, sub-10-nm line patterns were obtained in thin films of PS-*b*-PG_FMs with PGMA contents less than 11 mol% in a highly reliable and reproducible manner. Although, the line patterns of PS-*b*-PMMA with $L_0$ of less than 22 nm are

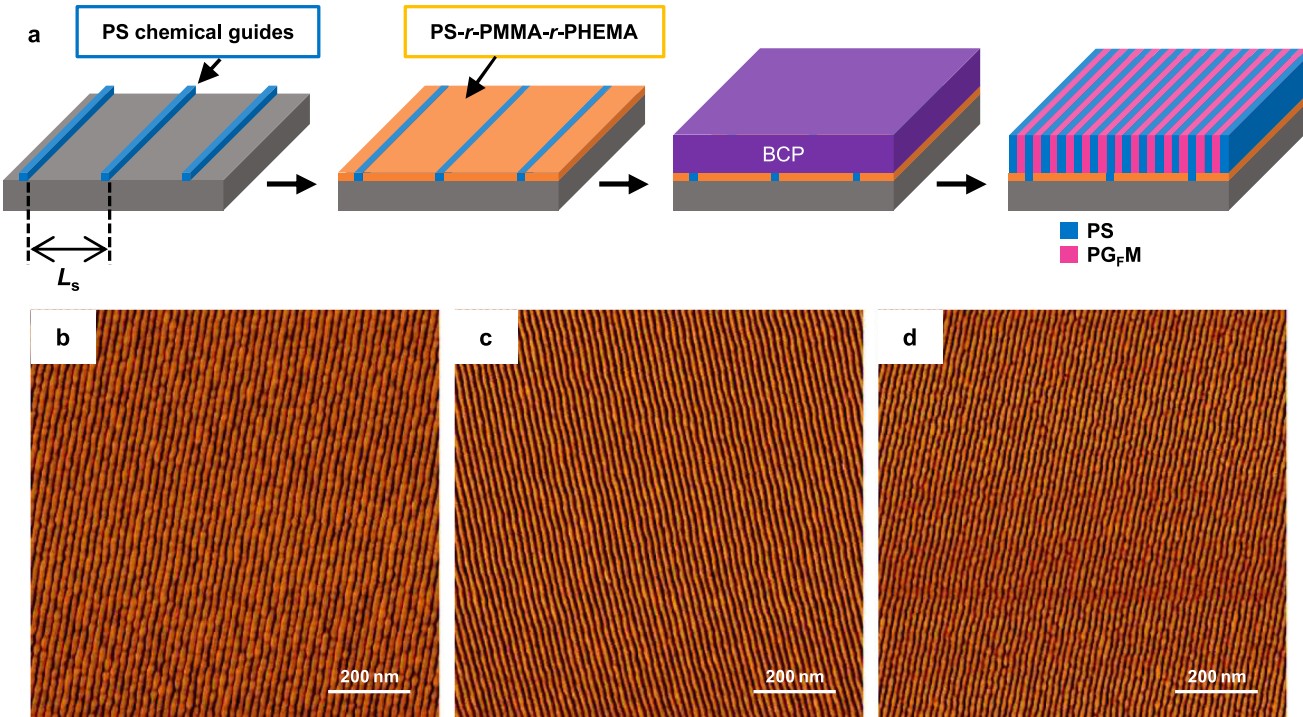

**Fig. 7 | Fabricating sub-20-nm line patterns by DSA using PS-*b*-PG$_F$M.**
**a** Schematic of the DSA process using a PS-*b*-PG$_F$M on a chemically patterned Si substrate. AFM phase images of a **b** PS-*b*-PG$_F$M19-23 film on an NL35-modified DSA substrate ($L_s$ = 90 nm) after annealing at 240 °C for 5 min, **c** PS-*b*-PG$_F$M19-10 film on an NL38-modified DSA substrate ($L_s$ = 84 nm) after annealing at 230 °C for 5 min, and **d** PS-*b*-PG$_F$M18-11 film on an NL38-modified DSA substrate ($L_s$ = 90 nm) after annealing at 230 °C for 5 min. All thin films are 19-nm thick and were etched using O$_2$ plasma for 10 s prior to AFM.

not able to be applied to lithography because of weak contrast between PS and PMMA domains[29], the fine line pattern of PS-*b*-PG$_F$M with $L_0$ of 15.1 nm has potential to act as a template for pattern transfer to semiconductor substrates.

## Discussion

In this study, we used sequential living anionic polymerization and the thiol–epoxy reaction to synthesize PS-*b*-PG$_F$M block copolymers with various percentages of PGMA as PS-*b*-PMMA derivatives with high Flory–Huggins interaction parameters ($\chi$). We investigated the relationship between PGMA content, BCP phase-separation behavior, and the surface affinity of the PG$_F$M segment, and optimized the primary structure of PS-*b*-PG$_F$M to prepare an ultrafine perpendicular lamellar structure through simple thermal annealing. PS-*b*-PG$_F$M bulk films were analyzed by SAXS and TEM, which revealed well-ordered lamellae with a minimum *d*-spacing of 12.4 nm, which is markedly smaller than the lower limit of the pattern size achieved using unmodified PS-*b*-PMMA. Effective $\chi$ ($\chi_{eff}$) values of 0.110, 0.133, and 0.142 were determined for PS-*b*-PG$_F$Ms with 10, 22, and 30 mol% PGMA at 200 °C, respectively, based on the random-phase approximation with a reference volume of 118 Å$^3$ used to normalize the degree of polymerization over the volume. More importantly, we found that the PS-*b*-PG$_F$Ms formed perpendicular lamellar structures with lamellar domains ($L_0$) of less than 20 nm in the thin-film state after simple thermal annealing. Tilted SEM revealed vertical lamellar domains inside the thin films. In particular, PS-*b*-PG$_F$M10-22 exhibited the smallest $L_0$ of 12.3 nm in this study. Furthermore, PS-*b*-PG$_F$M polymers with less than 23 mol% PGMA were compatible with the DSA process, and 7.6-nm-wide line patterns along PS chemical guides were successfully obtained. Introducing 10–23 mol% 2,2,2-trifluoroethyl groups into the PMMA blocks of PS-*b*-PMMA increased the $\chi_{eff}$ value without significantly changing the SFE of the PMMA segment. PS-*b*-PG$_F$M BCPs are promising templates for use in BCP lithography because they produce fine patterns in a similar DSA process to that used for PS-*b*-PMMA and have the potential to outperform PS-*b*-PMMA. Studies aimed at optimizing the pattern-transfer processes using line patterns in PS-*b*-PG$_F$M thin films as templates will be investigated in the future.

## Methods

### Materials

Styrene (>99%), DPE (>98%), GMA (>95%), and MMA (>99.8%) were purchased from the Tokyo Chemical Industry Co., Ltd. GMA was first passed through an activated alumina column, while styrene, GMA, and MMA were distilled over calcium hydride (CaH$_2$) (≥90.0%) under reduced pressure. The purified styrene, GMA, and MMA were degassed and further distilled over *n*-butyl-*sec*-butylmagnesium (0.7 M in *n*-hexane), CaH$_2$, and trioctylaluminum (25 wt% in *n*-hexane), respectively. DPE was degassed and distilled over *n*-butyllithium (1.6 M in *n*-hexane). All other reagents and solvents were purchased from the following sources and used as received: Kanto Chemical Co., Inc., Nacalai Tesque, Inc., Sigma–Aldrich, Tokyo Chemical Industry Co., Ltd., GODO Co., Ltd., Fujifilm Wako Pure Chemical Co., or Koso Chemical Co., Ltd. Cross-linkable polystyrene mats (X-PS) was provided by the Tokyo Ohka Kogyo Co., Ltd.

### Characterization

$^1$H and $^{13}$C NMR spectra of the polymers were recorded on a JEOL 400 MHz instrument at ambient temperature using deuterated chloroform as the solvent. The number-averaged and weight-averaged molecular weights ($M_n$ and $M_w$, respectively) were determined by SEC against PS standards using a Shimadzu Prominence501 system equipped with a refractive index detector and two LF-804 columns (Shodex), with THF as eluent at 40 °C. Fourier-transform infrared (FT-IR) spectra were obtained using a JASCO FT/IR-4100 Fourier-transform spectrophotometer with potassium bromide as the matrix. Thermogravimetric analysis (TGA) was conducted using an

EXSTAR7000 series TG/DTA7300 instrument (Hitachi High Tech) between 30 and 550 °C at 10 °C min⁻¹. Differential scanning calorimetry (DSC) was carried out using an EXSTAR7000 series DSC7020 instrument (Hitachi High Tech) at 5 °C min⁻¹. Water and diiodomethane contact angles were measured using a Kyowa DM-501YH system. SAXS experiments were carried out using a Bulker NanoSTAR instrument (50 kV/50 mA) with a VANTEC-500 detector (camera length: 1 m) to investigate the microphase-separated structure in the bulk state and the temperature dependence of $\chi_{eff}$. TEM (H-7650, Hitachi High Tech) was used to examine 80-nm-thick thin samples prepared by room-temperature ultramicrotomy (EM UC7, Leica) and stained with RuO₄. Thin film samples were prepared using a spin coater (1H-D7, Mikasa), and their thicknesses were measured using a Filmetrics F20-EXR film thickness measurement instrument. AFM (NanoWizard Ultra Speed A, JPK) was used to observe the surface architecture of annealed BCP thin films. Tilted cross sections of BCP thin films were observed using field-emission SEM (SU9000, Hitachi High Tech).

### Synthesis of PS-*b*-PGM10-22 by living anionic polymerization

All polymerization procedures were conducted after purging with argon. LiCl (183 mg, 4.32 mmol, >99.95%) was added to a 300-mL Schlenk flask and dried under reduced pressure while heated with a heat gun. After cooling to room temperature, THF (90 mL, >99.5%) was added to the flask and then cooled to −78 °C using a cooling bath. A solution of *sec*-BuLi in cyclohexane and *n*-hexane (1.2 M) was added until the color changed to yellow, and the mixture was stirred for 10 min. The flask was removed from the cooling bath and allowed to warm to room temperature until the solution became colorless, after which it was re-cooled to −78 °C, and *sec*-BuLi in cyclohexane and *n*-hexane (0.720 mL, 0.600 mmol) was added to initiate the reaction. Styrene (3.60 mL, 31.3 mmol) was added and the orange mixture was stirred for 30 min. DPE (0.500 mL, 2.86 mmol) was then added and the deep-red solution was stirred for a further 30 min. A mixture of GMA (0.658 mL, 5.00 mmol) and MMA (2.13 mL, 20.0 mmol) was then added and the mixture was stirred for 30 min, during which time it became colorless. Degassed methanol (20 mL, >99%) was added to the flask to prepare proton-terminated PS-*b*-PGM10-22. The polymer precipitated from MeOH and was collected by filtration. The small amount of PS homopolymer produced during polymerization was removed by Soxhlet extraction with *n*-hexane. The remaining solid was dissolved in THF, precipitated from MeOH, collected by filtration, and dried under reduced pressure at 40 °C to obtain PS-*b*-PGM10-22 as a white powder (5.09 g, 85% yield). ¹H NMR (400 MHz, CDCl₃, δ, ppm): 0.56–1.17 (α-C*H*₃, PGMA; α-C*H*₃, PMMA), 1.43 (-C*H*₂-CH-, PS), 1.82 (-C*H*₂-C*H*-, PS; -C*H*₂-C(CH₃)-, PGMA; -C*H*₂-C(CH₃)-, PMMA), 2.64 (-CH₂-CH(C*H*₂)-O-, PGMA), 2.86 (-CH₂-CH(C*H*₂)-O-, PGMA), 3.23 (-CH₂-C*H*(CH₂)-O-, PGMA), 3.60 (-O-C*H*₃, PMMA), 3.75–3.85 (-C*H*₂-CH(CH₂)-O-, PGMA), 4.26–4.38 (-C*H*₂-C H(CH₂)-O-, PGMA), 6.26–6.84 (*o*-aromatic, PS), 6.84–7.24 (*m*, *p*-aromatic, PS). ¹³C NMR (100 MHz, CDCl₃, δ, ppm): 16.6, 18.8, 40.4, 44.6, 44.9, 48.9, 49.1, 51.9, 54.3, 65.9, 125.6, 125.7, 127.4, 127.5, 127.7, 128.0, 128.1, 128.3, 145.1, 145.4, 145.7, 146.1, 177.0, 177.2, 177.9, 178.2. IR (KBr, *v*, cm⁻¹): 3447, 3103, 3083, 3060, 3026, 3000, 2948, 2925, 2849, 1943, 1867, 1799, 1733, 1631, 1602, 1493, 1453, 1438, 1388, 1268, 1241, 1192, 1149, 1065, 1027, 989, 967, 908, 845, 756, 699, 542.

### Synthesis of PS-*b*-PG<sub>F</sub>M10-22 by the thiol−epoxy reaction

General procedure for functionalizing PGM random copolymers and PS-*b*-PGM BCPs using PS-*b*-PGM10-22 as an example: The reaction vessel was charged with PS-*b*-PGM10-22 (0.308 g, 0.0305 mmol), 2,2,2-trifluoroethanethiol (0.0370 mL, 0.416 mmol; 1.5 mol·equiv. of PGMA repeating units, 95%), and 1 wt% aqueous LiOH·H₂O solution (0.0580 g, 0.0138 mmol; 0.05 mol·equiv. of pre PGMA repeating units). The reactants were dissolved in THF (3 mL; 1 mL per 100 mg of PS-*b*-PGM10-22) at ambient temperature. The resulting solution was then

stirred at 40 °C for 3 h. The polymer was precipitated from MeOH/water and collected by filtration. The solid was dissolved in THF, precipitated from MeOH/water, and collected by filtration to remove residual thiol. The resulting solid was dried under reduced pressure at room temperature to obtain PS-*b*-PG<sub>F</sub>M10-22 as a white powder (0.321 g, 94% yield). ¹H NMR (400 MHz, CDCl₃, δ, ppm): 0.53–1.18 (α-C*H*₃, PGMA <sub>F</sub>; α-C*H*₃, PMMA), 1.43 (-C*H*₂-CH-, PS), 1.82 (-C*H*₂-C*H*-, PS; -C*H*₂-C(CH₃)-, PGMA<sub>F</sub>; -C*H*₂-C(CH₃)-, PMMA), 2.84 (-CH(OH)-C*H*₂-S-, PGMA<sub>F</sub>), 3.27 (-S-C*H*₂-CF₃, PGMA<sub>F</sub>), 3.60 (-O-C*H*₃, PMMA), 4.07 (-(C = O)O-C*H*₂-C*H*(OH)-, PGMA<sub>F</sub>), 6.30–6.84 (*o*-aromatic, PS), 6.84–7.24 (*m*, *p*-aromatic, PS). ¹³C NMR (100 MHz, CDCl₃, δ, ppm): 16.5, 17.3, 18.8, 34.3, 34.6, 34.9, 35.3, 36.2, 40.3, 44.6, 44.9, 52.0, 54.4, 67.8, 69.1, 124.6, 125.6, 125.7, 127.4, 127.5, 127.7, 128.0, 145.2, 145.4, 145.8, 146.3, 177.1, 177.3, 178.0, 178.3. IR (KBr, *v*, cm⁻¹): 3504, 3103, 3082, 3060, 3026, 3000, 2947, 2925, 2848, 1940, 1863, 1799, 1731, 1601, 1493, 1453, 1434, 1385, 1313, 1271, 1244, 1193, 1151, 1125, 1084, 1024, 985, 963, 905, 841, 756, 699, 636.

### Preparing bulk films

Bulk films were prepared by slowly evaporating 10 wt% BCP solutions in THF at 25 °C. The prepared BCP films were annealed at 200 °C for 24 h under reduced pressure.

### Preparing thin films

Silicon wafers were cut into 1 × 1 or 1.5 × 1.5 cm pieces and wiped with toluene-soaked Kim Wipes. The wafers were then sequentially sonicated in acetone, ethanol (>99.5%), and toluene (>99%) (1 min each), dried in a stream of nitrogen, and further dried by heating to 100 °C. Random copolymer layers were prepared on the clean wafers by spin-coating 1 wt% PS-*r*-PMMA-*r*-PHEMA solutions in propylene glycol 1-monomethyl ether 2-acetate (PGMEA) (>98.0%) at 3000 rpm for 60 s, followed by crosslinking at 250 °C for 5 min under nitrogen. Subsequent rinsing with PGMEA removed the un-crosslinked random copolymers to yield PS-*r*-PMMA-*r*-PHEMA layers approximately 5–7-nm thick that neutralize the silicon wafers. These modified substrates were sufficiently dried by heating at 100 °C for 1 min. Films were then fabricated by spin-coating BCP solutions of given concentrations in PGMEA at various spin rates for 60 s in order to obtain thin films of various thicknesses. These BCP thin films were first heated to 90 °C for 1 min to remove residual PGMEA and then annealed at 200 °C for 30 min under nitrogen.

### DSA

SiN films with a thickness of 13 nm were deposited on Si substrates and crosslinked X-PS patterns were fabricated on the substrates (DSA substrates). The DSA substrates were obtained from imec and used as received. The DSA substrates were sufficiently dried by heating at 100 °C. Random copolymer layers were prepared by spin-coating 1 wt% PS-*r*-PMMA-*r*-PHEMA solutions in PGMEA at 1500 rpm for 30 s onto DSA substrates, followed by crosslinking at 250 °C for 5 min in air. Subsequent rinsing with 7:3 (v/v) PGME/PGMEA removed un-crosslinked random copolymers to yield PS-*r*-PMMA-*r*-PHEMA layers that neutralized the substrates. The modified substrates were sufficiently dried by heating at 100 °C for 1 min. BCP solutions of specific concentrations in PGMEA were then spin-coated at 2500 rpm for 30 s to produce 19-nm-thick BCP films that were heated at 90 °C for 1 min to remove residual PGMEA and then annealed at a given temperature for 5 min under nitrogen. The PG<sub>F</sub>M segments were finally removed by subjecting the thin films to oxygen-plasma etching.

## Data availability

All data are available in the main text or supplementary information. The data that support the findings of this study are available from the corresponding author on request.

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

## Acknowledgements

The authors thank Dr. Hyo Seon Suh and Dr. Lander Verstraete of the Interuniversity Microelectronics Centre (imec) for providing the DSA substrates. The authors also thank Ryohei Kikuchi of the Tokyo Institute of Technology Materials Analysis Division, Open Facility Center for TEM and SEM support. This work was supported by JSPS KAKENHI Grant Number 20H02785 and 24H00052. S.M. was supported by JST SPRING, Grant Number JPMJSP2106.

## Author contributions

T.H. and S.M. conceived the idea and initiated the project. S.M. conducted the experiments from synthesis and characterization of all polymer samples. T.S., T.D., and K.S. discussed the experimental conditions of DSA. T.H., Y.N., and K.H. provided technical guidance and useful suggestions on the material designs, characterization, and clear explanations of experiments. S.M., K.H., Y.N., and T.H. co-wrote the manuscript. All authors commented on the manuscript. T.H. supervised the whole project.

## Competing interests

T.D. and T.H. are inventors on a patent application (number 7213495, Japan), dealing with the use of PS-*b*-PG$_F$M with 10–90 mol% PGMA$_F$ units in the PG$_F$M. The remaining authors declare no competing interests.
