## [Peer Review File · Nature Communications]

REVIEWER COMMENTS

Reviewer #1 (Remarks to the Author):

In their manuscript, Maekawa and colleagues present a significant advance in the directed self-assembly of block copolymer lamellae on chemically patterned substrates, resulting in line patterns with pitches as small as 12 nm. The key innovation lies in their novel synthetic approach, which involves post-functionalizing polystyrene-block-[poly(glycidyl methacrylate)-random-poly(methyl methacrylate)] (PS-b-(PGMA-r-PMMA) or PS-b-PGM) block copolymers with thiols, specifically 2,2,2-trifluoroethanethiol. This method increases the Flory-Huggins interaction parameters (χ) by 3.5-4.6 times compared to conventional PS-b-PMMA, which is a critical factor in achieving sub-10 nm period sizes through block copolymer self-assembly. The study relies on standard characterization techniques such as small-angle X-ray scattering (SAXS) and transmission electron microscopy (TEM) to confirm the lamellar structures and spacings. In addition, the authors use disordered-SAXS profile fitting based on the established Leibler mean-field theory to calculate the Flory-Huggins interaction parameters.

The authors use well-established protocols to fabricate patterned substrates, specifically sparse line patterns with chemical contrast. These substrates were prepared by modifying silicon wafers with a silicon nitride (SiN) layer and using PS chemical guides for directed self-assembly. Adding PS-r-PMMA-r-PHEMA random copolymers to these substrates enables the directed self-assembly of a series of PS-b-PGFM block copolymers, ultimately forming 12 nm line patterns.

Overall, this work is compelling and represents a notable contribution to the directed self-assembly of block copolymers for next-generation lithography. The synthesis of block copolymers with enhanced interaction parameters is a critical step towards achieving sub-10 nm feature sizes through self-assembly. The data presented in this manuscript are of high quality. The study covers a comprehensive spectrum, including structural characterization, determination of interaction parameters, and successful fabrication of small pitch line patterns through directed self-assembly.

However, there is a notable omission in the manuscript regarding a recent comparison with other approaches to generating small-pitch line patterns via block copolymer self-assembly with high interaction parameters. While the authors cite some relevant work, including references such as <https://doi.org/10.1021/acsnano.5b02613> and <https://doi.org/10.1021/acs.macromol.8b01325>, they do not directly compare their results to these studies. In addition, they fail to mention other relevant studies from the literature, such as <https://doi.org/10.1021/acsmacrolett.8b00293>, <https://doi.org/10.1021/acs.macromol.9b00174>, <https://doi.org/10.2494/photopolymer.28.659>, <https://doi.org/10.1021/acs.macromol.8b01409>, <https://doi.org/10.1021/nl2016224>, and <https://doi.org/10.1021/acsnano.7b02698>. It is essential to directly compare interaction parameters and pitch sizes between the block copolymers used in this study and those in other relevant publications to place the results in context and allow the editor to assess their suitability for publication in Nature Communications.

In conclusion, the manuscript by Maekawa et al. is a very promising and well-executed study on the directed self-assembly of block copolymers yielding sub-10 nm patterns. To strengthen the scientific impact and relevance of the paper, it is crucial to include a more comprehensive comparative analysis of their results with other recent findings in the literature. This addition will increase the scientific rigor of the manuscript and aid in its evaluation for publication in Nature Communications.

Reviewer #2 (Remarks to the Author):

This article claims the smallest perpendicular lamellae assembled from block polymers which had a 12.3 nm domain size. As apparent in review articles elsewhere (10.1021/acsmacrolett.5b00472) and the introduction here, this is a very crowded field with widespread similar claims where this article appears to slightly advance the state of the art.

It must be pointed out that while this polymer-lithography field is scientifically interesting, it does not appear to be technologically relevant. Since 2022 production lithography at TSMC has reached 3 nm feature sizes and did so with far lower defect densities and complete design flexibility. This disconnect from rational utility significantly tempers the enthusiasm and possible impact of this work.

The article is not particularly written for a general nature-style audience, and reads rather more like a Macromolecules article. It is also not written as a communication as it feels exceedingly long. While I do not feel this article is a good fit for Nature Comm, I can offer suggestions for how to improve it for future review elsewhere:

1) The parabolic trajectory of X with mol% PGMA (Fig 4b) should be explained in terms of theory/models. Seems like a missed opportunity to reflect more deeply upon the data.

2) The SAXS experiments to measure X were not sufficiently described. How did the authors measure the instrumental broadening factor (beam smearing)? How did the authors account for this instrumental affect when calculating X values?

3) The chemical sketches (Fig 2) were too small to be read easily. Changing the layout would allow more room for the figure to be readable.

Reviewer #3 (Remarks to the Author):

This manuscript reported route to increase the χ_{eff} by introducing 2,2,2-trifluoroethyl groups into the PMMA segments in PS-b-PMMA. Two kinds of PS-b-PMMA derivatives with higher χ values are obtained, which maintain the surface and interfacial properties of PS-b-PMMA. The work is important and publishable, but the following issues would be suggested to address.

1. In introduction, the first and second paragraphs are repetitive, and need to be trimmed. Besides, the presentation of the background focuses entirely on the method of increasing the χ value, and needs to be expanded.
2. Why does PS-b-PGM10-22 exhibit a disordered structure? Try to state the reason if there is one.
3. The positioning of numbers and text in the Tables needs to be adjusted, which are not aesthetically pleasing enough.
4. Some ticker labels should be removed in some Figures such as Fig.4.
5. In some sentences, the present and past tense are mixed, please correct it.
6. In the introduction part, the following publications are recommended to cite:

Adv. Mater. 2019, 1806254

Angewandte Chemie International Edition, 2023, 62, e202306261

Angewandte Chemie International Edition, 2023, 62, e202304420

Angewandte Chemie International Edition, 2022, 61, e202209038

Point to Point Response to the Reviewers

Manuscript ID; NCOMMS-23-40950A

We sincerely appreciate the editor and reviewers' valuable and insightful comments on our manuscript. We have revised and improved our manuscript as suggested. In the following, we provide a point-to-point response to every question raised. We have highlighted portions of the manuscript that have been updated in response to the reviewers' comments.

Reviewer #1 (Remarks to the Author):

In their manuscript, Maekawa and colleagues present a significant advance in the directed self-assembly of block copolymer lamellae on chemically patterned substrates, resulting in line patterns with pitches as small as 12 nm. The key innovation lies in their novel synthetic approach, which involves post-functionalizing polystyrene-block-[poly(glycidyl methacrylate)-random-poly(methyl methacrylate)] (PS-b-(PGMA-r-PMMA) or PS-b-PGM) block copolymers with thiols, specifically 2,2,2-trifluoroethanethiol. This method increases the Flory-Huggins interaction parameters (χ) by 3.5-4.6 times compared to conventional PS-b-PMMA, which is a critical factor in achieving sub-10 nm period sizes through block copolymer self-assembly. The study relies on standard characterization techniques such as small-angle X-ray scattering (SAXS) and transmission electron microscopy (TEM) to confirm the lamellar structures and spacings. In addition, the authors use disordered-SAXS profile fitting based on the established Leibler mean-field theory to calculate the Flory-Huggins interaction parameters.

The authors use well-established protocols to fabricate patterned substrates, specifically sparse line patterns with chemical contrast. These substrates were prepared by modifying silicon wafers with a silicon nitride (SiN) layer and using PS chemical guides for directed self-assembly. Adding PS-r-PMMA-r-PHEMA random copolymers to these substrates enables the directed self-assembly of a series of PS-b-PGFM block copolymers, ultimately forming 12 nm line patterns.

Overall, this work is compelling and represents a notable contribution to the directed self-assembly of block copolymers for next-generation lithography. The synthesis of block copolymers with enhanced interaction parameters is a critical step towards achieving sub-10 nm feature sizes through self-assembly. The data presented in this manuscript are of high quality. The study covers a comprehensive spectrum, including structural characterization, determination of interaction parameters, and successful fabrication of small pitch line patterns through directed self-assembly.

→**RESPONSE:** We thank the reviewer for these very positive comments on the work.

However, there is a notable omission in the manuscript regarding a recent comparison with other approaches to generating small-pitch line patterns via block copolymer self-assembly with high interaction parameters. While the authors cite some relevant work, including references such as <https://doi.org/10.1021/acsnano.5b02613> and <https://doi.org/10.1021/acs.macromol.8b01325>, they do not directly compare their results to these studies. In addition,

they fail to mention other relevant studies from the literature, such as <https://doi.org/10.1021/acsmacrolett.8b00293>, <https://doi.org/10.1021/acs.macromol.9b00174>, <https://doi.org/10.2494/photopolymer.28.659>, <https://doi.org/10.1021/acs.macromol.8b01409>, <https://doi.org/10.1021/nl2016224>, and <https://doi.org/10.1021/acsnano.7b02698>. It is essential to directly compare interaction parameters and pitch sizes between the block copolymers used in this study and those in other relevant publications to place the results in context and allow the editor to assess their suitability for publication in Nature Communications.

→**RESPONSE:** We thank the reviewer for these comments. We agree with the reviewer and have incorporated the suggestions in our manuscript.

1. We have re-cited <https://doi.org/10.1021/acsnano.5b02613> at the end of “Directed Self-Assembly” in order to compare the half-pitch of PS-*b*-PMMA and that of PS-*b*-PG_FM (line 258-261 on page 12, ref. 33).
2. To clarify that traditional high- χ BCPs such as <https://doi.org/10.1021/acs.macromol.8b01325> have difficulty forming perpendicular lamellae in thin films, the text has been changed from “However, the segment with the lower SFE selectively segregates to the surface, with parallel-oriented structures formed in polymer thin films.” to “However, the segment with the lower SFE selectively segregates to the surface, with parallel-oriented structures formed in most of the high- χ BCP thin films.”(line 47-48 on page 2).
3. By citing <https://doi.org/10.1021/acsnano.7b02698> and describing that top-coat is used in this earlier study, we clarify that this study has achieved DSA without top-coat (line 51-52 on page 2, ref. 43).
4. Periodic lengths of the lamellae (L_0) in high- χ BCPs obtained by thermal annealing have been added in the text to easily compare this study and earlier studies (line 59-62 and 67 on page 2-3).

In conclusion, the manuscript by Maekawa et al. is a very promising and well-executed study on the directed self-assembly of block copolymers yielding sub-10 nm patterns. To strengthen the scientific impact and relevance of the paper, it is crucial to include a more comprehensive comparative analysis of their results with other recent findings in the literature. This addition will increase the scientific rigor of the manuscript and aid in its evaluation for publication in Nature Communications.

→**RESPONSE:** We thank the reviewer for positive comments and their valuable feedback.

Reviewer #2 (Remarks to the Author):

This article claims the smallest perpendicular lamellae assembled from block polymers which had a 12.3 nm domain size. As apparent in review articles elsewhere (10.1021/acsmacrolett.5b00472) and the introduction here, this is a very crowded field with widespread similar claims where this article appears to slightly advance the state of the art.

It must be pointed out that while this polymer-lithography field is scientifically interesting, it does not appear to be technologically relevant. Since 2022 production lithography at TSMC has reached 3 nm feature sizes and did so with far lower defect densities and complete design flexibility. This disconnect from rational utility significantly tempers the enthusiasm and possible impact of this work.

→**RESPONSE:** We thank the reviewer for comments to improve our manuscript and the insightful feedback.

FinFET has been used for 3 nm process developed by TSMC in 2020. FinFET is a technology for wiring circuit patterns in three dimensions. The resulting patterns perform better than two-dimensional wiring patterns. The minimum line width of circuit patterns in the 3 nm process is about 12 nm, which is confusing. Therefore, there is a need to develop a technology to fabricate sub-10 nm features. The combination of DSA using high- χ BCP and EUV lithography is one of the next-generation lithography processes that will contribute to the fabrication of sub-10 nm features with low-defect and low-cost in the 2.1 nm process after 2025 (https://irds.ieee.org/images/files/pdf/2022/2022IRDS_Litho.pdf). According to IRDS2022, the target minimum line width by 2037 is 8 nm, and the line pattern with line width of 7.6 nm produced by DSA of PS-*b*-PG_FM meets this requirement. To make it easier to understand the above, the beginning of the introduction has been revised (line 23-29 on page 2).

The article is not particularly written for a general nature-style audience, and reads rather more like a Macromolecules article.

→**RESPONSE:** We have added a general introduction about semiconductor manufacturing at the beginning of the introduction (line 23-29 on page 2).

It is also not written as a communication as it feels exceedingly long.

→**RESPONSE:** We agree that this manuscript is not written as a communication. However, Nature communications is an article journal. This manuscript has been written to meet the journal's requirements.

While I do not feel this article is a good fit for Nature Comm, I can offer suggestions for how to improve it for future review elsewhere:

1) The parabolic trajectory of X with mol% PGMA (Fig 4b) should be explained in terms of theory/models. Seems like a missed opportunity to reflect more deeply upon the data.

→**RESPONSE:** We consider two reasons for the increase in the χ_{eff} value with increasing mol% of PGMA. The first reason is the increase in the repulsion between PS and PG_FM because of the trifluoroethyl group. Generally, the χ

values of fluorine-containing BCPs are higher than that of PS-*b*-PMMA. The second reason is that the hydrogen bonding caused by the hydroxyl groups increased the attractive interactions in the PG_FM block. It has been reported by Zhou et al. that increasing attractive interactions in a block contributes to increasing the χ_{eff} value. We have added the considerations to the main text (line 191-197 on page 10, ref. 38, 47, and 52).

2) The SAXS experiments to measure X were not sufficiently described. How did the authors measure the instrumental broadening factor (beam smearing)? How did the authors account for this instrumental affect when calculating X values?

→**RESPONSE:** Instrumental broadening factor was not considered in this study. However, the χ_{eff} value of PS-*b*-PMMA estimated using same instrument reproduced the results of earlier studies well. Additionally, the primary peaks of the SAXS profile used for the analysis were extremely broad and the plots in figure 4a were all straight lines. Therefore, in this study, we consider that the influence of instrumental broadening factor is small. The consideration has been added to the caption of Figure 4 (line 203-206 on page 10).

3) The chemical sketches (Fig 2) were too small to be read easily. Changing the layout would allow more room for the figure to be readable.

→**RESPONSE:** The layout of Figure 2 has been modified to make the chemical sketches easier to understand (Fig. 2, page 5).

Reviewer #3 (Remarks to the Author):

This manuscript reported route to increase the χ_{eff} by introducing 2,2,2-trifluoroethyl groups into the PMMA segments in PS-*b*-PMMA. Two kinds of PS-*b*-PMMA derivatives with higher χ values are obtained, which maintain the surface and interfacial properties of PS-*b*-PMMA. The work is important and publishable, but the following issues would be suggested to address.

→**RESPONSE:** We thank the reviewer for these very positive comments on the work.

1. In introduction, the first and second paragraphs are repetitive, and need to be trimmed.

→**RESPONSE:** The introduction has been revised to avoid duplication of meaning (line 30-69 on page 2-3).

Besides, the presentation of the background focuses entirely on the method of increasing the χ value, and needs to be expanded.

→**RESPONSE:** As you mentioned, the introduction focused on increasing the χ value to reduce the feature size. In this study, we designed a higher- χ BCP that aims to reduce the pitch size while retaining the advantages of PS-*b*-PMMA. The introduction has been revised to make the intent of molecular design easier to understand (line 39-40, 51-52, and 54-56 on page 2, line 69 on page 3).

2. Why does PS-*b*-PGM10-22 exhibit a disordered structure? Try to state the reason if there is one.

→**RESPONSE:** We consider that the disordered structure was formed in PS-*b*-PGM10-22 because the χN value of the polymer was less than 10.5. The χN value of PS-*b*-PG_FM10-22, in which the smallest structure was formed in this study, was estimated to be 15.8. The χN value of PS-*b*-PGM10-22 is expected to be even smaller. Although this point is not clear from the results of this study alone, it does not affect the thesis or results of this study (line 155-156 on page 8).

3. The positioning of numbers and text in the Tables needs to be adjusted, which are not aesthetically pleasing enough.

→**RESPONSE:** The tables have been revised in accordance with the regulations of Nature Communications (Table 1 and 2 on page 5-7).

4. Some ticker labels should be removed in some Figures such as Fig.4.

→**RESPONSE:** The labels in Figure 4b have been removed (Fig. 4 on page 10).

5. In some sentences, the present and past tense are mixed, please correct it.

→**RESPONSE:** We have corrected all of these errors and reviewed the manuscript fully.

6. In the introduction part, the following publications are recommended to cite:

Adv. Mater. 2019, 1806254

Angewandte Chemie International Edition, 2023, 62, e202306261

Angewandte Chemie International Edition, 2023, 62, e202304420

Angewandte Chemie International Edition, 2022, 61, e202209038

→**RESPONSE:** We have cited Adv. Mater. 2019, 1806254 in the introduction. Due to strict limitations of the journal, we cannot include all these references in the main text (line 31-32 on page 2, ref. 3).

Again, thank you for giving us the opportunity to strengthen our manuscript with your valuable comments and queries. We have worked hard to incorporate your feedback and hope that these revisions persuade you to accept our submission.

REVIEWERS' COMMENTS

Reviewer #2 (Remarks to the Author):

The authors have satisfactorily conveyed a possibility of technological relevance and the introductory materials were improved.

Regarding the first question of the reported parabolic trajectory for X: The authors repeated the reasons for the monotonic changes in X but did not add intellectual discourse about the specific parabolic shape. Curiously this trajectory is inconsistent with the expectation for X-approximation based on Hildebrand solubility parameters. Is this X-trend consistent with the expectations of continuum thermodynamics? The authors should discuss the parabolic shape further.

The second question regarding SAXS measurement of X indicated that the authors neglected instrumental broadening but felt encourage since a prior calculation of X they performed also neglected this attribute and resulted in a similar X value as reported elsewhere. While that is suggestive, it would be more proper to at least measure the instrumental broadening and report the value as a justification for neglecting this relatively simple correction. The correction is relatively simple to implement, often times accomplished by using the instrument broadening function to smear the RPA function during the fit.

Point to Point Response to the Reviewer

Manuscript ID: NCOMMS-23-40950B

We sincerely appreciate the editor and reviewer's valuable and insightful comments on our manuscript. We have revised and improved our manuscript as suggested. In the following, we provide a point-to-point response to every question raised. We have highlighted portions of the manuscript that have been updated in response to the reviewer's comments.

Reviewer #2 (Remarks to the Author):

The authors have satisfactorily conveyed a possibility of technological relevance and the introductory materials were improved.

→**RESPONSE:** We thank the reviewer for the very positive comment on the introduction.

Regarding the first question of the reported parabolic trajectory for X: The authors repeated the reasons for the monotonic changes in X but did not add intellectual discourse about the specific parabolic shape. Curiously this trajectory is inconsistent with the expectation for X-approximation based on Hildebrand solubility parameters. Is this X-trend consistent with the expectations of continuum thermodynamics? The authors should discuss the parabolic shape further.

→**RESPONSE:** We agree that our manuscript was not sufficient to explain the parabolic trend of the χ_{eff} value of PS-*b*-PG_FM. It has already been reported that the intra-block interactions contribute to the sudden increase in χ_{eff} values of A-*b*-(B-*r*-C) copolymers (10.1021/acs.macromol.6b01382) (10.1021/acs.macromol.1c00472).

To clarify our consideration of the parabolic trends, we have added discussion and references to the part of χ parameter in the manuscript (line 172-184 on page 5, ref. 49, 58, 66, 67, and 68). In order to comply with the journal regulation, the references in the main text have been revised to include a total of 70 references.

The second question regarding SAXS measurement of X indicated that the authors neglected instrumental broadening but felt encourage since a prior calculation of X they performed also neglected this attribute and resulted in a similar X value as reported elsewhere. While that is suggestive, it would be more proper to at least measure the instrumental broadening and report the value as a justification for neglecting this relatively simple correction. The correction is relatively simple to implement, often times accomplished by using the instrument broadening function to smear the RPA function during the fit.

→**RESPONSE:** We acknowledge the reviewer's suggestion for additional experiments to determine the instrumental broadening factor. While we understand the importance of suggested experiments, conducting the additional experiments would be estimated to be several months because of technical problems. Although we recognize the value these experiments could add, we believe that the absence of this data does not fundamentally alter the major claims and conclusions of our paper.

To address the concerns raised, we have added the discussion of the χ_{eff} values of PS-*b*-PMMA estimated using our SAXS instrument and that of estimated by previous studies in Supplementary Discussion (line 407-409 on page 32, Supplementary References 4-8, in the Supplementary Information). In addition, references, which describe the χ_{eff} values of PS-*b*-PMMA, have been added in the caption of Figure 4 (line 572 on page 19, ref. 28 and 70, in the main manuscript). In order to describe the certainty of the estimated χ_{eff} values of PS-*b*-PG_FMs, we have added Supplementary Table 10, which shows the estimated χN values and higher-order structures of PS-*b*-PG_FMs in bulk, and the discussion of the relationship between the critical χN value for order-disorder transition and observed higher-order structures (line 405 and 409-414 on page 32 in Supplementary Information).

We hope this approach adequately compensates for the lack of direct experimental data in the current context and plant to explore this aspect in greater detail in future studies.

Again, thank you for giving us the opportunity to strengthen our manuscript with your valuable comments. We have worked hard to incorporate your feedback and hope that these revisions persuade you to accept our submission.